# Characterization of Cytokines and Proliferation Marker Ki67 in Chronic Rhinosinusitis with Nasal Polyps: A Pilot Study

**DOI:** 10.3390/medicina57060607

**Published:** 2021-06-11

**Authors:** Rudolfs Janis Viksne, Gunta Sumeraga, Mara Pilmane

**Affiliations:** 1Department of Otorhinolaryngology, Pauls Stradins Clinical University Hospital, Pilsonu Street 13, LV-1002 Riga, Latvia; Gunta.Sumeraga@rsu.lv; 2Institute of Anatomy and Anthropology, Riga Stradins University, Kronvalda Boulevard 9, LV-1010 Riga, Latvia; Mara.Pilmane@rsu.lv

**Keywords:** cytokines, proliferation marker, nasal polyps, rhinosinusitis

## Abstract

*Background and Objectives:* Chronic rhinosinusitis (CRS) is a condition that affects as much as 10.9% of the population and, along with presence of nasal polyps, is associated with significant morbidity and decreased quality of life. Studies on molecular pathways that have been activated in nasal polyp tissue are mainly based on cytokine concentration detection. Therefore, our aim is to investigate the complex appearance, relative distribution and interlinks of IL-1, IL-4, IL-6, IL-7, IL-8, IL-10, IL-12 and Ki 67 in chronic rhinosinusitis with nasal polyps (CRSwNP) affected human nasal mucosa. *Materials and Methods*: Samples of nasal polyps were obtained from 12 patients with previously diagnosed CRSwNP and no prior surgery. Control group consisted of samples from 17 otherwise healthy individuals with isolated nasal septum deviation. Tissues were stained for IL-1, IL-4, IL-6, IL-7, IL-8, IL-10, IL-12 and Ki67 immunohistochemically. Non-parametric statistic, Mann–Whitney U test and Spearman’s rank correlation coefficient were used. *Results*: All factors, except connective tissue cytokine IL-10 and proliferation marker Ki-67, had increased presence in connective tissue and decreased presence in epithelium of nasal polyps when compared to controls. Very strong and strong positive correlations between factors were observed. *Conclusions*: Decreased appearance of IL-1α, IL-4, IL-6, IL-7, IL-8, IL-10, IL-12 positive structures in the nasal epithelium with selective increase of IL-1α and IL-12 in nasal subepithelial connective tissue characterize the cytokine endotype with dysfunctional epithelial barrier and local stimulation of immune response in the connective tissue in case of chronic rhinosinusitis with polyps. Decrease of IL-6 in both—epithelium and connective tissue with strong correlation between it and IL-7 and IL-10 in connective tissue suggests significant stimulation of this regulatory cytokine and, possibly, the important role in pathogenesis of the development in nasal polyps. Correlations between Ki67 and cytokines indicate possible involvement of IL-4, IL-7 and IL-12 in regulation of cellular proliferation.

## 1. Introduction

Chronic rhinosinusitis (CRS) is a condition that affects as much as 10.9% of the population of Europe [1]. It is defined as a presence of two or more of the following symptoms: nasal congestion or blockage, nasal discharge, facial pain, pressure, reduction or loss of smell, one of which should be either nasal blockage or nasal discharge for more than twelve weeks [2]. CRS is related with a significant impact on patients’ quality of life, affecting their work efficiency, social interactions, and daily living. CRS patients report on disrupted sleep, leading to fatigue, as well as feeling depressed, sad and upset about their condition [3,4]. About 25% to 30% of patients with CRS have nasal polyps [5]. Nasal polyps are inflammatory masses growing from the mucosa of paranasal sinuses [6]. Chronic rhinosinusitis with polyps (CRSwNP) is associated with more significant morbidity and decreased quality of life [5]. In the pediatric population chronic rhinosinusitis tends to manifest fewer cases of nasal polyps and neutrophilic inflammation is more common, whereas in adults eosinophilic inflammation is more pronounced [7]. In the geriatric population age-related changes, thinning of nasal mucosa and decreased mucociliary clearance are considered additional aspects that affect the pathogenesis of chronic rhinosinusitis. Furthermore, due to comorbidities and use of blood thinners geriatric rhinosinusitis presents challenges for both surgical and pharmacological management [8].

Recently research has shifted from identifying a causative agent to recognizing molecular pathways that have been activated in inflamed tissue [2]. Many authors have undertaken the task of determining the molecular mechanisms of CRS [9,10,11,12]. Their studies have focused on analysis of mucus cytokines, various clinical factors, and presence of nasal polyps and asthma. Other studies have focused on analyzing specific factor causality in pathogenesis of CRSwNP such as IL-6 and IL-8 [13,14].

Patients with chronic rhinosinusitis often also suffer from bronchial asthma which is an argument for a systemic inflammatory disease rather than an isolated problem [15]. Since bronchial asthma as well as chronic rhinosinusitis has been associated with type 2 inflammatory response, identification of cytokines in CRS affected nasal mucosa is beneficial in understanding chronic rhinosinusitis and asthma as a complex systemic process.

Identification of underlying molecular pathways in CRSwNP has potential benefits for the development of specific treatment options and reduces the need for surgery. Yet it is agreed that further research is required as there has been no improvement in endoscopic and accompanying pharmacological polyp control in the last decade [2,16].Thus the above-mentioned leads the interleukin research, firstly, to the morphopathogenesis of polyp development, and, despite already developed studies, no work looks on the complex research of cytokines in case of nasal polyps. We believe that a dysregulation of cytokine balance and their complex interactions in CRSwNP affected mucosa could be directly responsible for formation of polyps, as established by various authors in the last decade [9,10,12]. Evaluation of cytokines in both epithelial and subepithelial connective tissue is of potential benefit as it can help characterize the function of epithelial barrier, since it is established previously as defective in cases of nasal polyps [17].

Interleukin 1 (IL-1) is an important mediator of innate immunity and inflammation. IL-1 family consists of seven ligands with agonist activity (most notably IL-1α and IL-1β), three receptor antagonists and an anti-inflammatory cytokine IL-37. IL-1α precursor is present in various types of cells throughout the body, including epithelial cells, endothelial cells and upper respiratory tract cells [18]. IL-1α is expressed under normal conditions, it does not depend on proteolysis for bioactivity so, if a cell dies, it is released in its active form. [19]. IL-1α is designated as a key alarmin and therefore is worth examining in the context of CRS to evaluate tissue damage in nasal polyps [20].

Interleukin-4 (IL-4) is produced by T helper cells, basophils, natural killer T cells, mast cells and eosinophils. IL-4 takes a form of preformed peptide in eosinophils and basophils so it could be released during inflammation response. IL-4 stimulates various immune cell receptors and enhances antigen presentation [21]. IL-4 is also produced by group 2 innate lymphoid cells and is responsible for adaptive and innate signaling in all of these cell types [22]. Interleukin-4 (IL-4) along with interleukin-13 (IL-13), are key factors in initiation and continuing of type 2 inflammatory response and have a role in various atopic diseases [23]. Type 2 inflammatory response has been associated with CRS, and so IL-4 detection is a factor of interest in CRS cytokine research [2].

Interleukin 6 (IL-6) is a pleiotropic pro-inflammatory cytokine and can be found in almost every organ system [24]. IL-6 is produced mainly by mononuclear phagocytic cells, but can be produced by B and T lymphocytes, eosinophils, granulocytes, mast cells, endothelial cells, keratinocytes and bone marrow cells [25]. Considering the previously established IL-6 role in CRSwNP, it is a factor of interest when evaluating nasal polyps.

The main function of interleukin 7 (IL-7) is to regulate the development and survival of T cells. IL-7 is known to be secreted continuously by stromal cells and under normal circumstances therefore ensuring survival of T lymphocytes, but in cases of lymphopenia elevated IL-7 levels are observed [26]. IL-7 is a factor of interest due to its role in T lymphocyte survival and potentially can characterize their role in CRSwNP.

IL-8 is a mediator of leukocyte recruitment and neovascularization [13]. IL-8 is secreted by various types of cells in response to an inflammation [27]. IL-8 has a potent effect in airway inflammation both acute and chronic. In a small study it was observed that tissue of nasal polyposis showed increased IL-8 presence thus supporting hypothesis that local production of IL-8 could be an important factor in recruiting leukocytes [13]. IL-8 is established as a contributing factor in CRSwNP and could be a basic inducer for neutrophilic infiltration of the tissue.

Interleukin 10 (IL-10) is an anti-inflammatory cytokine that prevents excessive inflammation. It can be secreted by any immune cells to downregulate inflammatory effects [28]. IL-10 detection is necessary for evaluation of anti-inflammatory responses in nasal polyps.

Another interleukin, interleukin 12 (IL-12), is a pro-inflammatory cytokine and is produced by dendritic cells and phagocytes when responding to pathogens. It causes the secretion of interferon-γ (IFN-γ) and facilitates differentiation of T helper 1 cells (Th1) [29]. IL-12 detection could help evaluate involvement of adaptive immune responses in nasal polyps. Due to epithelial and sub-epithelial connective tissue proliferation in CRSwNP, additionally, proliferation marker Ki 67 requests the evaluation in order to determine its possible connection to cytokines in tissue remodeling process. Ki 67 is widely known as a proliferation marker for human tumor cells. It has a role in inter-phase and during mitosis—it aids in forming perichromosomal layer—a coating for condensed chromosomes [30]. Because of Ki 67 activity during mitosis and its inactivity during cell resting periods, it has been a useful tool for grading of proliferation in various cancer types [31]. Ki 67 has also been observed in tissue with severe inflammation [32].

On the basis of the abovementioned facts, the aim of this work was to investigate the complex appearance, relative distribution and interlinks of IL-1, IL-4, IL-6, IL-7, IL-8, IL-10, IL-12 and Ki 67 in CRSwNP-affected human nasal mucosa.

## 2. Materials and Methods

### 2.1. Subject Characteristics

Samples of nasal polyps were obtained from 12 patients with previously diagnosed chronic rhinosinusitis with nasal polyps (CRSwNP) and no prior nasal polyp surgery. The study group consisted of seven male and five female patients with an average age of 52.4 years. Four of the patients reported a previously diagnosed allergy, and seven patients were diagnosed with bronchial asthma. Patients with previously diagnosed CRSwNP were included in the study group, but exclusion criteria were coagulopathies, immunodeficiencies or exacerbation of CRS symptoms 2 weeks prior to surgery. Control group consisted of inferior turbinate mucosa samples from 17 otherwise healthy individuals with an average age of 39 years, who were diagnosed with isolated nasal septum deviation. Control group exclusion criteria were coagulopathies, immunodeficiencies or previously diagnosed CRS. Research was approved by ethics committee of Riga Stradins University (6-1/10/59. 26 October 2020). A written consent was obtained from every patient. Nature of the study was explained, and patients voluntarily agreed to take part in the research.

### 2.2. Immunohistochemical Analysis

Samples were initially collected in a mixture of 2% formaldehyde and 0.2% picric acid in 0.1 M phosphate buffer (pH 7.2) for up to 72 h. After that they were rinsed in Tyrode buffer (content: NaCl, KCl, CaCl_2_·2H_2_O, MgCl_2_·6H_2_O, NaHCO_3_, NaH_2_PO_4_·H_2_O, glucose) containing 10% saccharose for 12 h and then embedded into paraffin. Three micrometers thin sections were cut and then stained with hematoxylin and eosin for routine morphological evaluation. Biotin-streptavidin biochemical method was used for immunohistochemistry (IMH) to detect: Ki-67 (1325506A, 1:100, Cell Marque, Rocklin, CA, USA), IL-1 α (orb308737, 1:100, Biorbyt, Cambridge, UK), IL-4 (orb10908, 1:100, Biorbyt, UK), IL-6 (sc-130326, 1:100, Santa Cruz Biotechnology Inc., Dallas, TX, USA), IL-7 (orb13506, 1:100, Biorbyt, Cambridge, UK), IL-8 (orb39299, 1:100, Biorbyt), IL-10 (250713, 1:100, BioSite, Täby, Sweden), IL-12 (orb10894, 1:100, Biorbyt). The slides were analyzed with light microscopy. The results were evaluated by using semi-quantitative counting method for the appearance of positively stained cells in the visual field. Structures in the visual field were categorized as follows: no positive structures in the visual field were labelled as 0, rare positive structures were labelled with 0/+, few positive structures: +, few to moderate number of positive structures in the visual field: +/++, moderate number of positive structures in the visual field: ++, moderate to numerous positive structures in the visual field: ++/+++, numerous positive cells in the visual field: +++, numerous to abundant structures in the visual field: +++/++++ and abundant positive structures in the visual field was labelled as ++++ [33].

### 2.3. Statistical Analysis

Data analysis was made using SPSS software (Version 26.0 IBM Corp. Chicago, IL, USA). The results from semi-quantitative evaluation were transformed into numerical form as follows: 0 equals to 0, 0/+ equals to 0.5, + equals to 1, +/++ equals to 1.5, ++ equals to 2, ++/+++ equals to 2.5, +++ equals to 3. Spearman’s rank correlation coefficient and the Mann–Whitney U test (nonparametric test to compare outcomes between two independent groups) were used to assess correlations between factors in patients with nasal polyps as well as significant differences between them and the control group. Spearman’s rank correlation coefficient was used for nonparametric measure of rank correlation—statistical dependence of ranking between two variables. Spearman’s rank correlation coefficient with a value of ≥0.81 meant a very strong relationship between two variables, 0.61–0.80 meant a strong relationship, 0.41–0.60 a moderate relationship, 0.21–0.40 a weak relationship, and 0.01–0.20 no relationship [34]. Mann–Whitney U test was used as a nonparametric test to compare outcomes between two independent groups. *p*-value was calculated to assess significance between two samples. If the *p*-value was less than 0.05, it rejected the null hypothesis that the samples were of equal value [35].

## 3. Results

Routinely stained nasal polyp samples with hematoxylin and eosin (HE) demonstrated oedema of subepithelial connective tissue, infiltration of leukocytes with a predominance of eosinophils as well as thickening of the basal membrane (Figure 1a). The epithelium consisted of stratified squamous and respiratory epithelium; however, glands were present at variable degree (Figure 1b). Some samples of nasal polyps had large cysts forming in residual glandular tissue (Figure 1c). Proliferation of basal epithelial cells was also seen (Figure 1d).

Immunohistochemistry data revealed that IL-1α showed moderate to numerous positive structures in epithelium of control samples, but nasal polyp samples only had few to moderate positive epitheliocytes. When observing connective tissue, in contrast to epithelium, control samples showed occasional number of IL-1α positive structures and nasal polyp samples had moderate amount of positive cells (Figure 2a,b).

Numerous IL-4 positive cells were detected in epithelium and few to moderate number in connective tissue of controls (Figure 2c). In nasal polyp samples there were few to moderate IL-4 positive epitheliocytes, but moderate positive structures were seen in connective tissue (Figure 2d).

When evaluating IL-6, control group patients had numerous positive epitheliocytes, but connective tissue only showed few positive cells (Figure 3a). In contrast, nasal polyps revealed few to moderate IL-6 positive structures in epithelium and moderate to numerous in connective tissues (Figure 3b).

Abundant IL-7 positive structures were observed in epithelium of controls, but connective tissue only showed few to moderate number of positive cells (Figure 3c). Nasal polyp samples demonstrated few to moderate IL-7 positive epitheliocytes, and numerous positive cells in connective tissue were found (Figure 3d).

Abundant IL-8 positive cells were detected in control sample epithelium, unlike nasal polyp sample epithelium that only showed few positive structures. When observing IL-8 in connective tissue, control samples showed few positive structures and nasal polyp samples had few to moderate number of positive cells (Figure 4a,b).

Abundant IL-10 positive structures were detected in epithelium of controls and moderate number of IL-10 positive structures—in the connective tissue (Figure 4c). Nasal polyp samples revealed few to moderate positive IL-10 structures in epithelium and moderate number of positive structures in connective tissue (Figure 4d).

Numerous IL-12 positive structures were seen in epithelium of control samples and few positive structures in connective tissue (Figure 5a). Few IL-12 positive structures were seen in epithelium of nasal polyps and numerous positive structures in connective tissue (Figure 5b).

Few to moderate number of Ki67 positive structures were seen in the epithelium of the control sample and occasional positive structures in connective tissue (Figure 5c). Similarly, a few Ki67 positive cells were seen in epithelial as well as in connective tissue of nasal polyps (Figure 5c).

The amount of IL-1α positive structures in nasal polyps was significantly decreased in epithelium (*p* = 0.001) and increased in connective tissue (*p* < 0.001) in comparison to control samples. IL-4 positive structures in nasal polyps were significantly decreased in epithelium (*p* < 0.001) and increased in the connective tissue (*p* = 0.014) in comparison to control samples. Amount of IL-6 positive structures in nasal polyps was significantly decreased in epithelium (*p* < 0.001) and increased in the connective tissue (*p* < 0.001) in comparison to control samples. IL-7 positive structures in nasal polyps were significantly decreased in epithelium (*p* < 0.001) and increased in the connective tissue (*p* = 0.006) in comparison to control samples.

IL-8 positive structures in nasal polyps were significantly decreased in epithelium (*p* < 0.001) and increased in connective tissue (*p* = 0.028) in comparison to control samples. IL-10 positive structures in nasal polyps were significantly decreased in epithelium (*p* < 0.001) in comparison to control samples, but there were no significant differences between amounts of positive structures in the connective tissue. IL-12 positive structures in nasal polyps were significantly decreased in epithelium (*p* = 0.002) and increased in the connective tissue (*p* < 0.001) in comparison to control samples. There were no significant differences in Ki67 positive structures in nasal polyp epithelium and the connective tissue when compared to control samples (Table 1).

We found a very strong positive correlation between the number of epithelial IL-8 positive structures and epithelial IL-10 (R = 0.930, *p* < 0.001), connective tissue IL-10 (R = 0.926, *p* < 0.001), epithelial IL-7 (R = 0.901, *p* < 0.001) and positive structures of connective tissue IL-8 (R = 0.883, *p* = 0.001). A very strong positive correlation was observed between epithelial IL-7 positive structures and connective tissue IL-8 (R = 0.884, *p* = 0.001), connective tissue IL-10 (R = 0.866, *p* = 0.001), epithelial IL-10 (R = 0.819, *p* = 0.004) positive structures. A very strong positive correlation was also observed between positive structures of epithelial IL-10 and the connective tissue IL-10 (R = 0.901, *p* < 0.001), as well as between connective tissue IL-8 and connective tissue IL-10 (R = 0.877, *p* < 0.001) positive structures (Figure 6).

A strong correlation was observed between positive structures of epithelial IL-6 and epithelial IL-8 (R = 0.761, *p* = 0.011), epithelial IL-10 (R = 0.694, *p* = 0.026) and the connective tissue IL-10 (R = 0.665, *p* = 0.036) positive structures. A strong correlation was seen between positive structures of connective tissue Ki67 and connective tissue IL-12 (R = 0.779, *p* = 0.003), connective tissue IL-7 (R = 0.623, *p* = 0.031) and connective tissue IL-4 (R = 0.620, *p* = 0.032) positive structures. Connective tissue IL-8 positive structures strongly correlated with epithelial IL-10 (R = 0.735, *p* = 0.016), connective tissue IL-6 (R = 0.722, *p* = 0.008) and connective tissue IL-1α (R = 0.678, *p* = 0.015) positive structures. There was a strong positive correlation between connective tissue IL-4 and epithelial Ki67 (R = 0.640, *p* = 0.046) and connective tissue IL-6 (R = 0.625, *p* = 0.033) positive structures. We observed a strong positive correlation between positive structures of connective tissue IL-1α and epithelial IL-1α (R = 0.726, *p* = 0.017) and epithelial IL-7 (R = 0.716, *p* = 0.020) positive structures, as well as between connective tissue IL-6 and connective tissue IL-7 positive structures (R = 0.733, *p* = 0.007) (Figure 7). In our study we did not observe weak correlations among analyzed factors.

## 4. Discussion

Nasal polyps of our patients demonstrated variable morphological features in the tissue as the different types of epithelia, thickened basal membrane, subepithelial oedema and inflammation with inflammatory cells, various distribution of the glands, and even cysts. These findings commonly respond to the already described structure of polyps’ affected nasal tissue and are supposed to be nonspecific, as it is a well-known fact that structure of nasal polyps differs from that of normal nasal mucosa. Subepithelial regions in nasal polyps consist mostly of connective tissue infiltrated with various lymphocytes, also neutrophils and eosinophils. Presence of glands in nasal polyp tissue seems scarce compared to normal nasal mucosa. Respiratory epithelium of nasal mucosa is often replaced by stratified squamous epithelium in nasal polyps as dysplasia is accompanying epithelial and subepithelial proliferation of cells. Thickening of the epithelial basal membrane was often observed. It is already established as a characteristic finding in CRS and is associated with cases that are hard to manage [36,37]. Epithelial dysplasia and thickening of basal epithelial membrane would suggest that epithelial barrier in CRSwNP is becoming defective as it has been established by previous studies [17].

When comparing nasal polyp samples with normal mucosa from control patients, it was apparent that IL-1α, IL-4, IL-6, IL-7, IL-8, IL-10, IL-12 positive structures were significantly decreased in epithelium of the polyps. Thus we suppose the abovementioned polyp epithelium to be characterized by a complex cytokine distribution endotype, which is unique and proves the changed epithelial functions in case of chronic rhinosinusitis with polyps. Among all cytokines, IL-1α and IL-12 showed higher presence, but IL-6—was decreased in the connective tissue of polypous nasal mucosa, showing, probably, these cytokines as most changeable in the case of this disease. Interestingly, when using a variable chronical rhinosinusitis endotype analysis, IL-1β was most often the choice. One author found an endotype with solely increased IL-1β and smaller percentage of nasal polyps and asthma [11]. In contrast, another study found increased levels of IL-1β along with IL-6 and IL-8 and a high percentage of nasal polyps [9]. A different study analyzed patients with chronic rhinosinusitis with polyps only, and one of the clusters showed an increase in IL-1β [12], but the other author did not obtain statistically significant evidence of IL-1β value being higher in at least one of the analyzed clusters [10]. Thus, our data about the decreased expression of IL-1α just proves the probably insignificant role of this IL in the inflammation of nasal polyps or indicates the possible compensation for the expression of other IL-1 type (IL-1β) in case of the disease.

IL-4 showed a decrease of positive cells in epithelium of nasal polyp patients and an increase in subepithelial connective tissue. IL-4 has been associated with type 2 immune mechanism in nasal polyps [23], and suggested to be a reason for a change in epithelial permeability in nasal polyps. Furthermore, this change in epithelial barrier is believed to increase allergen exposure, tissue oedema and nasal discharge in nasal polyps [38]. Since IL-4 is responsible for the regulation of initial differentiation of naive T helper cell type 0 (Th0) lymphocytes into the Th2 lymphocytes [39], we suggest that its function is becoming less active in epithelial cells, but is elevated in subepithelial tissue as a consequence to a failing epithelial barrier. IL-4 has been observed in many CRSwNP-related studies and has been found to have a high concentration in tissue samples with nasal polyps [9,40,41]. One study characterized expression patterns of several type 2 inflammatory cytokines including IL-4 and found that only patients with chronic rhinosinusitis with polyps and allergic fungal rhinosinusitis demonstrated an increase in IL-4 gene expression [41].

Like other factors, IL-6 was found significantly less in epithelium of our patients, but significantly more in connective tissue of nasal polyps when compared to controls. All over the world there is very different data describing the IL-6 distribution in cluster analysis of chronic rhinosinusitis. For example, in a Chinese population, Wei et al. described in 2018 distinctive chronic rhinosinusitis with polyps endotype that includes elevated Il-6 and IL-8 [12]. Liao et al. also described in 2018 high IL-6 concentration for similar patients [9]. In North America, Turner et al. identified two clusters of patients with elevated IL-6, both of which were characterized with high inflammation and high prevalence of nasal polyps and asthma [11]. In Europe, Tomassen et al. have described ten distinct clusters or endotypes, where five out of ten clusters showed an elevated IL-6 levels, while the correlation between asthma, nasal polyposis and IL-6 level was variable [10]. Commonly, IL-6 could contribute to exaggerated epithelial reaction and growth of nasal polyps [14], which, probably, is not the most characteristic feature in our patients. However, we suggest the indistinct functions of this regulatory cytokine in polypous epithelium but increase in significance of IL-6 in the connective tissue, probably, due to the stimulation of all other cytokines/local defense reactions in the tissue. This our suggestion is proved by the facts that IL-6 in epithelium of our patients has revealed a strong correlation with epithelial IL-8 and IL-10, as well as with connective tissue IL-10. IL-6 in connective tissue also had a strong correlation with connective tissue IL-7, IL-4, and IL-8. Relationship between IL-6 and IL-8 has been established by Wei et al., Turner et al. and Tomassen et al. as a specific inflammation type in CRS [10,11,12].

IL-7 expression was decreased in our CRSwNP, while its appearance in connective tissue was practically similar in both—patients and controls. Epithelial IL-7 has shown a very strong correlation with both epithelial and connective tissue IL-10 and IL-8. IL-7 has not been used extensively in sample studies of CRS tissue and mostly was present in patients with nasal polyps [9,11]. Also IL-7 and other cytokine interactions have not been evaluated in CRSwNP, thus our findings about this cytokine are evaluated as new in the field. IL-7 regulates the development and survival of T lymphocytes [26]. Detectable IL-7 tissue levels are regulated by CD4 and CD8 T cells that consume IL-7 after expressing receptor IL-7R to ensure their survival [42]. Based on above-mentioned we connect the decrease of IL-7 in epithelium and persistence in connective tissue of nasal polyps mainly with the role of this cytokine in the common local adaptive immune responses of connective tissue.

Our findings with the decreased IL-8 appearance in epithelium of CRSwNP and similar expression in underlying connective tissue of patients and controls suggest the specific endotype in case of nasal polyps with imbalance of epithelial cell functions. The mucosal epithelial cells are believed to be the most important source of IL-8 in chronic rhinosinusitis [43]. It has also been hypothesized that locally produced IL-8 could contribute to progression of CRSwNP [13]. However, data are incomplete and contrary. So, in Europe, the nasal endotype cluster analysis performed by Tomassen et al. revealed only five out of ten clusters showing elevated IL-8 levels, but correlation between asthma, nasal polyposis and IL-8 levels was variable [10]. In a Chinese population, Wei et al. found IL-8 in one endotype from analyzed patients with chronic rhinosinusitis with polyps [12]. While we detected a decrease of IL-8 in epithelium, probably, there exist more than one endotype characteristic for the CRSwNP patients, and larger analysis of epigenetic factors is needed in the future research. Additionally, human neutrophils exhibit a strong directional migration towards increasing IL-8 [44], which seemingly was not the dominating in our patients. Finally, a very strong correlation between epithelial IL-8 and both epithelial and connective tissue IL-10 was observed indicating that an active opposing anti-inflammatory response takes place in CRSwNP.

Epithelial IL-10 was significantly decreased in nasal polyps when compared to controls. On the other hand, there were no differences between connective tissue IL-10, thus we suppose the impaired local epithelial immune functions basing on unchanged function in the connective tissue. Interestingly, an increased IL-10 has been previously associated with CRSwNP [9,10,11,45]. One study showed a downregulation of CD8+T-cells via IL-10 as a novel pathophysiologic mechanism of chronic rhinosinusitis [46]. Controversially, it is also concluded that impaired IL-10 production of M2 macrophages may facilitate sustained eosinophilic inflammation in CRSwNP [47]. Haruna et al. found that exposure to *Staphylococcus aureus* enterotoxin B and consequentially impaired IL-10 production in nasal polyps may exacerbate pathophysiology of eosinophilic chronic rhinosinusitis [48].

Our patients showed a decrease of IL-12 expression in the epithelium with significant increase of IL-12 positive connective tissue cells. Thus, we speculate on the impaired role of this IL into the epithelium, but an increase of its significance in connective tissue. For example, it might be IL-12 function of differentiation of T helper 1 cells [29] that is reduced in epithelium and increased in connective tissue of nasal polyps, although controversial findings in nasal polyps have been described. In chronic rhinosinusitis endotype cluster analysis, made by Turner et al., elevated IL-12 level was found in one endotype with increased percentage of nasal polyps and asthma [11]. However, IL-12 was increased in one endotype that was characterized with a low count of difficult-to-treat cases of rhinosinusitis in a different study [9]. Decreased serum levels of IL-12 have been demonstrated in patients with chronic rhinosinusitis [49]. IL-12 expression appears to be also decreased in mucosa samples from allergic and non-allergic chronic rhinosinusitis patients [50].

Ki 67 was not significantly increased in nasal polyps when compared to controls, which indicates the lack of serious cellular proliferation. There is scarce information regarding the Ki 67 correlation with cytokines in general, let alone in CRSwNP. Our previous works revealed that a correlation between Ki 67 and cytokines IL-8 and IL-10 indicates the involvement of these cytokines in cell proliferation in cleft lip tissue [33]. In other studies, using Western blot and immunofluorescence a higher Ki 67 levels were observed in nasal polyps [51,52]. Yet another study found positive Ki 67 smears in all of their control patients as well as in 65% of nasal polyp samples [32]. Ki 67 presence in our samples more likely is an indicator of indistinct proliferation in persistent inflammation. Additionally, strong positive correlations were observed between connective tissue IL-4 and both epithelial and connective tissue Ki 67. This supports findings of Rha et al. where an increase of Ki 67 expression from sinonasal CD4+ T cells in presence of IL-4 and Il-13 was observed [53]. Connective tissue Ki 67 had a strong correlation also with connective tissue IL-12 and connective tissue IL-7. Controversially, one study shows that Ki 67 expression is actually reduced when IL-12 is introduced in the tissue of fibrosarcoma [54]. Based on above-mentioned we suggest that commonly correlations between Ki67 and cytokines may indicate possible involvement of IL-4, IL-7 and IL-12 in regulation of cell proliferation, which mainly is not changed in case of CRSwNP. Furthermore, IL-4 has been known to increase transforming growth factor (TGF)-β in nasal polyps and therefore could be responsible for stromal cell proliferation [55].

This study certainly presents limitations. Due to the relatively small sample size, we have focused mainly on the immunohistochemistry data while also other methods, for instance, additional tissue analysis by ELISA method to determine concentration of cytokines and comparison with our immunohistochemical findings is a valuable direction in the future. In the future we plan to involve more patients with CRSwNP with both primary polyps and recurrent polyps after previous surgeries as well as include patients with chronic rhinosinusitis without nasal polyps (CRSsNP) to detect tissue factors that are possibly responsible for the polyp morphopathogenesis and to reveal the age-related differences in their development. Additionally, in the future we aim to correlate our findings with clinical phenotypes of CRS but as for now—our study is limited to basic clinical sciences and understanding cytokine and proliferation marker interactions.

Expression of cytokines in nasal mucosa can be affected by the presence of pathogens in nasal cavity and sinuses. Indeed *Staphylococcus aureus* presence and released toxins are noted to damage epithelial structure in nasal mucosa. Furthermore dysfunctional immune responses in nasal polyps facilitate *S. aureus* entry into the tissue. Intracellular *S. aureus* can further induce inflammatory cytokines and facilitate continuous inflammation [56,57]. To further our research, microbiological testing for intracellular *S. aureus* would be necessary as to to understand its complex interactions with established cytokine patterns in CRSwNP. Locally produced IgE against tissue *S. aureus* could also be beneficial since it is found in increased levels in nasal polys. [58,59]. *S. aureus* and cytokine interactions could also be beneficial in understanding eosinophilic chronic rhinosinusitis, which is a subgroup of CRSwNP that is characterized by severe eosinophilic infiltration that might be caused by *S. aureus* [58].

## 5. Conclusions

Decreased appearance of IL-1α, IL-4, IL-6, IL-7, IL-8, IL-10, IL-12 positive structures in the nasal epithelium with selective increase of IL-1α and IL-12 in nasal subepithelial connective tissue characterize the cytokine endotype with dysfunctional epithelial barrier and local stimulation of immune response in the connective tissue in case of chronic rhinosinusitis with polyps.

Decrease of IL-6 in both—epithelium and connective tissue with strong correlation between it and IL-7 and IL-10 in connective tissue suggests significant stimulation of this regulatory cytokine and, possibly, the important role in pathogenesis of the development in nasal polyps.

Correlations between Ki67 and cytokines indicate possible involvement of IL-4, IL-7 and IL-12 in regulation of cellular proliferation.

## Figures and Tables

**Figure 1 medicina-57-00607-f001:**
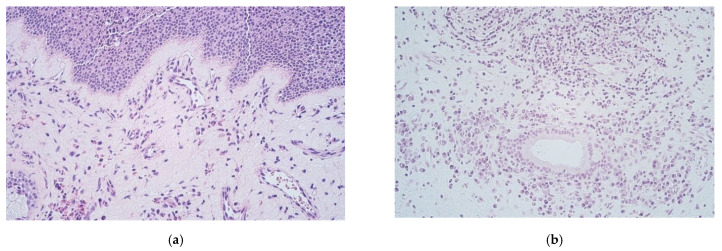
(**a**–**d**) Micrographs of routinely stained nasal polyps (**a**) 50-year-old male with CRSwNP. Sample of a nasal polyp showing stratified squamous epithelium, thickening of the basal membrane and subepithelial connective tissue infiltration of neutrophils and eosinophils. HE, X 250; (**b**) 60-year-old female with CRSwNP.Rare glandular tissue with periglandular leukocyte infiltration within the subepithelial connective tissue. HE, X 250; (**c**) 78-year-old female with CRSwNP. A fragment of a large cyst formed within subepithelial glandular tissue. HE, X 100; (**d**) 30-year-old female with CRSwNP. Note epithelial basal cell proliferation and subepithelial connective tissue infiltration with inflammatory cells in polyp tissue. HE, X 250.

**Figure 2 medicina-57-00607-f002:**
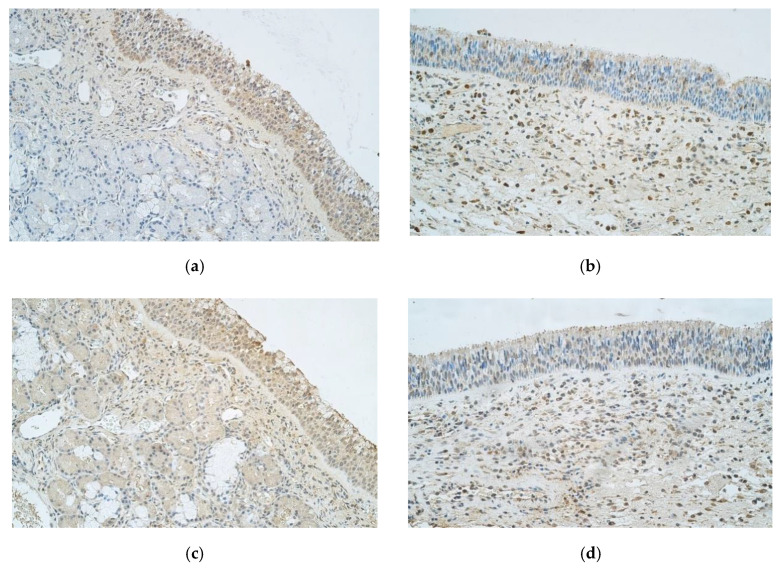
(**a**–**d**) Immunohistochemical micrographs of patients with nasal polyps and control subjects. (**a**) 42-year-old male with healthy nasal mucosa. Note moderate to numerous IL-1α positive structures in epithelium as well as occasional positive structures in connective tissue of the control sample of normal nasal mucosa and large amounts of subepithelial glandular tissue and. IL-1α IMH, X 250; (**b**) 33-year-old male with CRSwNP. Nasal polyp sample with few to moderate IL-1α positive structures in epithelium, moderate number of positive structures in connective tissue and numerous subepithelial connective tissue with edema. IL-1 α IMH, X 250; (**c**) 42-year-old male with healthy nasal mucosa. Numerous IL-4 positive structures in epithelium and few to moderate positive structures in subepithelial connective tissue of a control sample. IL-4 IMH, X 250; (**d**) 33-year-old male with CRSwNP. A sample of a nasal polyp showing few to moderate IL-4 positive structures in epithelium and moderate IL-4 positive structures in connective tissue. IL-4 IMH, X 250.

**Figure 3 medicina-57-00607-f003:**
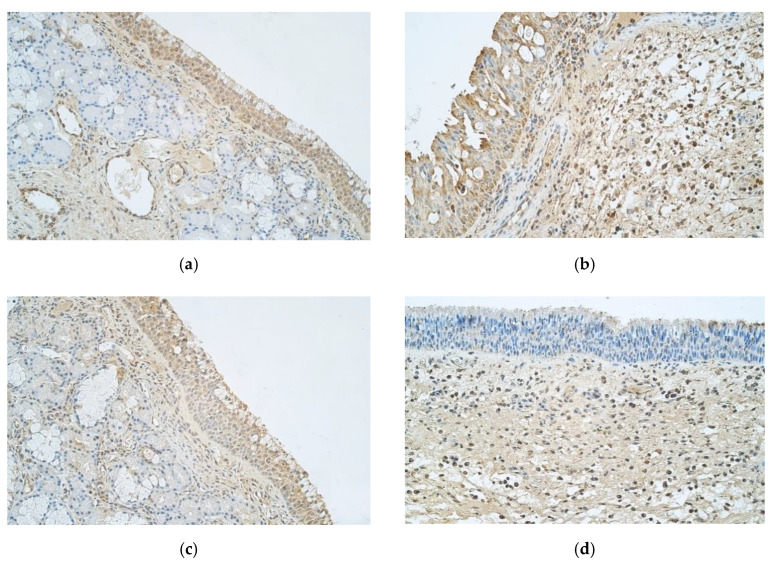
(**a**–**d**) Immunohistochemical micrographs of inpatients with nasal polyps and control subjects. (**a**) 42-year-old male with healthy nasal mucosa. Numerous IL-6 positive structures in the control sample epithelium and few positive structures in connective tissue. IL-6 IMH, X 250; (**b**) 30-year-old female with CRSwNP. Few to moderate IL-6 positive structures in epithelium and moderate to numerous IL-6 positive structures in connective tissue of a nasal polyp. IL-6 IMH, X 250; (**c**) 42-year-old male with healthy nasal mucosa. Numerous IL-7 positive structures in epithelium of a control sample with few to moderate positive structures in connective tissue. IL-7 IMH, X 250; (**d**) 33-year-old male with CRSwNP. Few to moderate positive structures in epithelium and moderate to numerous IL-7 positive structures in the connective tissue of a nasal polyp. IL-7 IMH, X 250.

**Figure 4 medicina-57-00607-f004:**
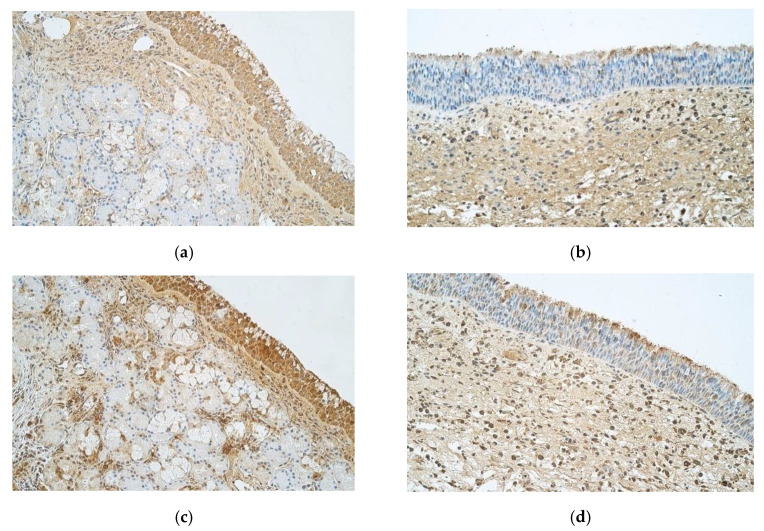
(**a**–**d**) immunohistochemical micrographs of inpatients with nasal polyps and control subjects. (**a**) 42-year-old male with healthy nasal mucosa. Note numerous to abundance of IL-8 positive structures in epithelium of the control sample and few positive structures in the connective tissue. IL-8 IMH, X 250. (**b**) 33-year-old male. Nasal polyp sample has few IL-8 positive structures in epithelium but few to moderate number of positive structures in connective tissue. IL-8 IMH, X 250. (**c**) 42-year-old male with healthy nasal mucosa. Abundance of IL-10 positive structures in the epithelium and moderate number of IL-10 positive structures in connective tissue of the control sample. IL-10 IMH, X 250. (**d**) 33-year-old male. A sample of a nasal polyp with few to moderate IL-10 positive structures in epithelium and moderate number of IL-10 positive structures in connective tissue. IL-10 IMH, X 250.

**Figure 5 medicina-57-00607-f005:**
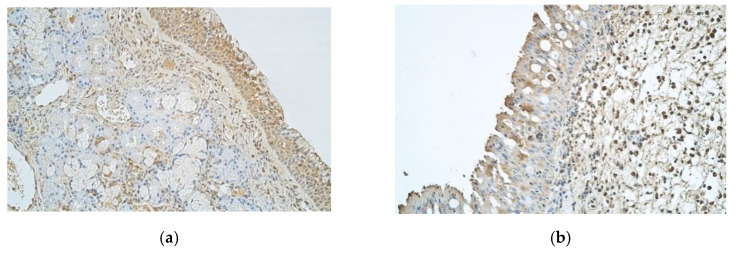
(**a**–**d**) Immunohistochemical micrographs of inpatients with nasal polyps and control subjects. (**a**) 42-year-old male with healthy nasal mucosa. Numerous IL-12 positive structures in the epithelium of the control sample, but few positive structures in the connective tissue. IL-12 IMH, X 250. (**b**) 30-year-old female with CRSwNP. Few IL-12 positive structures in epithelium and numerous positive structures in the connective tissue of a nasal polyp. IL-12 IMH, X 250. (**c**) 42-year-old male with healthy nasal mucosa. Few to moderate number of Ki67 positive structures in the epithelium and occasional positive structures in connective tissue of the control sample. Ki67 IMH, X 250. (**d**) 33-year-old male with CRSwNP. Few Ki67 positive structures in epithelium and connective tissue of a nasal polyp. Ki67 IMH, X 250.

**Figure 6 medicina-57-00607-f006:**
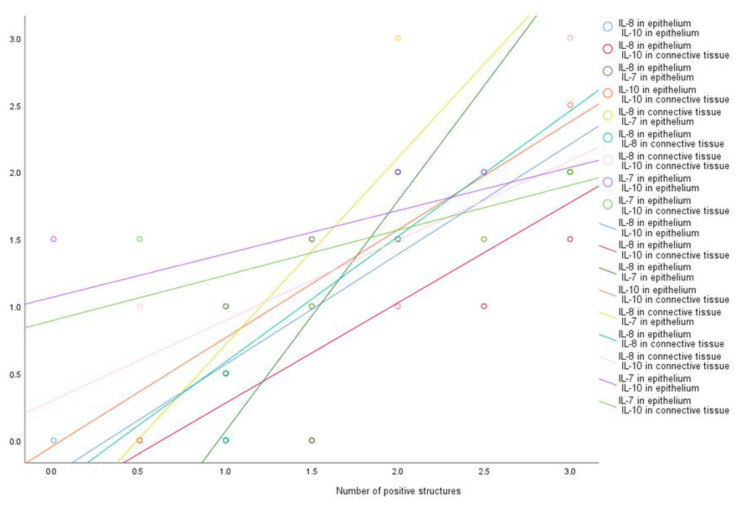
Very strong correlations among cytokines. Abbreviations: IL-1α —interleukin 1 alpha; IL-4—interleukin 4; IL-6—interleukin 6; IL-7—interleukin 7; IL-8—interleukin 8; IL-10—interleukin 10; IL-12—interleukin 12.

**Figure 7 medicina-57-00607-f007:**
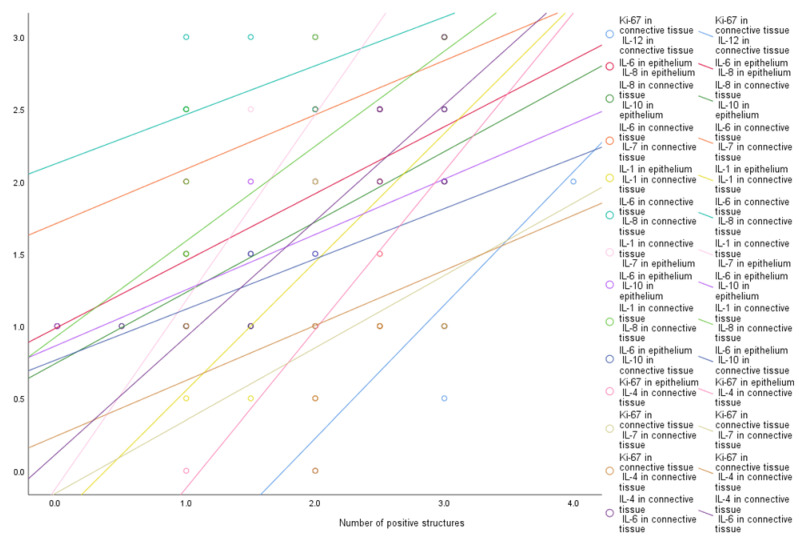
Strong correlations among cytokines and proliferation marker Ki 67. Abbreviations: Ki 67—proliferation marker; IL-1α —interleukin 1 alpha; IL-4—interleukin 4; IL-6—interleukin 6; IL-7—interleukin 7; IL-8—interleukin 8; IL-10—interleukin 10; IL-12—interleukin 12.

**Table 1 medicina-57-00607-t001:** Cytokines and proliferation marker Ki67 positive structures in samples of nasal polyps and significant differences between them and the control group.

Factors/Subjects	IL-1α		IL-4		IL-6		IL-7		IL-8		IL-10		IL-12		Ki 67	
	E	CT	E	CT	E	CT	E	CT	E	CT	E	CT	E	CT	E	CT
1	++	+++	++	++/+++	++	+++	++	+++	++	+++	++	+++	++/+++	+++	+	+
2	+++	+++	++	++/+++	++	++/+++	+	++	++	++	+++	+++	++	+++	+/++	+
3	++	++	+	++	+	+++	++	+++	+/++	++	++	+++	+	+++	++	+
4	0/+	++	+	++	++	+++	++	++	++	++	++/+++	+++	++/+++	+++	0/+	0/+
5		0/+		+/++		+		0/+		0		+		++		0/+
6	+	+/++	+	+	+	+/++	+	++/+++	0/+	+	+	+	++	+++	0	+
7	+/++	++/+++	+/++	+++	+	+++	+/++	+++	0	+	0	0/+	++	++/+++	++	+
8	++	++	+	++	+/++	++/+++	+/++	++	+	+	+/++	+/++	+++	++	0	0
9		+		+		++		+		+		+		+		0
10	+	+	+	++/+++	+	++/+++	+	++/+++	0	+	+	+/++	++	+++	++	+
11	0/+	+	+	++/+++	++	+++	+/++	+++	+	+/++	+/++	++/+++	++	+++	+/++	++/+++
12	0/+	+/++	++	++/+++	+/++	++/+++	+/++	++	+	+	++	++	+++	++++	++	++
Average	+/++	++	+/++	++	+/++	++/+++	+/++	++	+	+/++	+/++	++	++	+++	+	+
Control	++/+++	0/+	+++	+/++	+++	+	+++	+/++	+++/++++	+	++++	++	+++	+	+/++	0/+
Significant Difference (*p*-value)	0.001	<0.001	<0.001	0.014	<0.001	<0.001	<0.001	0.006	<0.001	0.028	<0.001	0.811	0.002	<0.001	0.309	0.073

*Abbreviations*: E—epithelium; CT—connective tissue; Ki 67—proliferation marker; IL-1α —interleukin 1 alpha; IL-4—interleukin 4; IL-6—interleukin 6; IL-7—interleukin 7; IL-8—interleukin 8; IL-10—interleukin 10; IL-12—interleukin 12; 0/+—occasional positive structures, +—few positive structures, +/++—few to moderate number of positive structures, ++—moderate number of positive structures, ++/+++—moderate to numerous positive structures, +++—numerous positive cells, +++/++++—numerous to abundance of structures, ++++—abundance of positive structures in the visual field.

## Data Availability

The data presented in this study are available on request from the corresponding author.

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
