# Peer review of "Characterization of Cytokines and Proliferation Marker Ki67 in Chronic Rhinosinusitis with Nasal Polyps: A Pilot Study"

_medicina, 2021, doi:10.3390/medicina57060607_

Round 1

Reviewer 1 Report

INTRODUCTION

- the authors should specify that CRS can occur at different ages, including the pediatric and geriatric population; moreover, the authors should also mention the association with other respiratory comorbidities (e.g. asthma, as raised later by the authors when discussing the cytokine environment), which partially account for and increase the impact of CRS on patients’ quality of life (Int J Immunopathol Pharmacol. 2010 Jan-Mar;23(1 Suppl):29-31; Otolaryngol Clin North Am. 2018 Aug;51(4):803-813).

- even though a clear and schematic background about the cytokines potentially involved in this immunopathological setting is useful, the introduction is definitely too long and must be shortened. The authors can provide a very short description of the main and general immunological role of each cytokine and postpone those specific concepts on CRSwNP to the discussion, only if appropriate and pertinent with the findings to be discussed.

- the last paragraph on Ki67 should be also summarized and the authors should postpone specific concepts and references to the discussion, if appropriate.

PATIENTS AND METHODS

- study and control groups are appropriately defined and the ethical aspects looks appropriate.

- overall, this section is clear enough.

RESULTS

- the histological pictures are interesting, but I do not think this is the best way to present these aspects and, importantly, there is no mention to the exact patients that the samples belong to. Moreover, the description should not be only qualitative, but some quantitative parameters (e.g. eosinophils? Others?) related to cell abundance and types could be provided.

- in general, this section should be reorganized in my opinion and would benefit of a structure in subsections describing patients’ characteristics (demographics, comorbidities, symptoms, etc.) general laboratory characteristics (which are both completely disregarded), histopathology, immunological aspects, etc. A table could be appropriate as well.

- as regards cytokines, again the authors should provide more explanation about their semi-quantitative assessment (“few”, “moderate”, “numerous” terms do not mean so much…a numerical range per field should be provided).

- in light of that, it is not very clear to me the statistical analysis.

- I think this section should be extensive revised, completed and reorganized.

- Are data on the local IgE production and microbiological data available? If yes, can the authors report that. Otherwise, some discussion on these aspects may be appropriate as well as the inclusion of missing information in the study limitations. Eosinophil-dominant inflammatory infiltration is found in most CRSwNP patients (70–90%), and is the expression of a Th2-polarized immune response and, thus, driven by the elevated production (e.g. IL-4, IL-5, IL-13) and IgE sensitization. Importantly, the role of some persistent microbial immunological stimulation (e.g. S. aureus) could be implicated, both in adults and children, which could also explain some important comorbidities (Toxins (Basel). 2020 Oct 28;12(11):678. doi: 10.3390/toxins12110678; Curr Opin Allergy Clin Immunol. 2020 Feb;20(1):1-8. doi: 10.1097/ACI.0000000000000588; Respir Med. 2018 Aug;141:94-99. doi: 10.1016/j.rmed.2018.06.016; Allergol Int. 2019 Oct;68(4):403-412. doi: 10.1016/j.alit.2019.07.002). Indeed, according to all these references, the discussion should be enriched as well as some study limitations should be disclosed.

- Indeed, in the materials and methods, the authors report that “Four of the patients reported a previously diagnosed allergy, and 7 patients were diagnosed with bronchial asthma”. By the way, this kind of information should be part of the results, after the authors revise and complete the results as recommended.

DISCUSSION

- I think the authors should begin the discussion by summarizing and highlighting the main findings, which then can be discussed point by point.

- again, I think that the discussion could be even shorter and more focused. In the current version, it appears quite dispersive.

- Importantly, there is no discussion of the several limitations of this study. The authors should actually add a paragraph on that. The previous comments and references provided about the results can be useful to further develop the discussion and describe the study limitations (including also the absence of an additional group of CRS without nasal polyps).

CONCLUSION

- “Correlations between Ki67 and cytokines indicate involvement of IL-4, IL-7 and IL-12 in regulation of cellular proliferation”. I am not sure this conclusion can be fully supported by the available data. I would invite the authors to explain better or revise this statement. This study can provide some interesting points (Laryngoscope. 2005 Apr;115(4):684-6. doi: 10.1097/01.mlg.0000161334.67977.5D).

REFERENCES

- to be revised, updated and completed according with the previous comments.

TABLES AND FIGURES

- the table and figure are graphically clear, but some specifications could be useful (see comments above)

- an additional table including demographic, clinical and laboratory characteristics of study and control groups.

Reviewer 2 Report

Dear Authors,

This is the interesting pilot study investigating the relationships among various cytokines.

Major concerns which should be addressed are below;

The reason why you decided to evaluate those cytokines in both of epithelial tissue and connective tissue. Those should be explained.

I just might miss the information, but I didn't get which tissue was taken for the analysis for controls.

How about weak correlations among cytokines in epithelium and connective tissue? 

As this is the pilot study, what you would suggest for future study should be noted.

Minor comments are below;

In materials and Methods, line 159-161 would be redundant. It was already mentioned above.

Table 2 would be redundant as it is also written in result section. I would recommend that it should be replaced scatter plot. Or tables for all results  including weak correlations would be nice.

Round 2

Reviewer 1 Report

The authors basically addressed all my previous comments. Importantly, they discussed the study limitations. No additional major comments.